# Digestibility Kinetics of Polyhydroxyalkanoate and Poly(butylene succinate-*co*-adipate) after In Vitro Fermentation in Rumen Fluid

**DOI:** 10.3390/polym14102103

**Published:** 2022-05-21

**Authors:** Hailey Galyon, Samuel Vibostok, Jane Duncan, Gonzalo Ferreira, Abby Whittington, Kirk Havens, Jason McDevitt, Rebecca Cockrum

**Affiliations:** 1Department of Dairy Science, Virginia Polytechnic Institute and State University, Blacksburg, VA 24061, USA; hailg98@vt.edu (H.G.); aduncan@vt.edu (J.D.); gonf@vt.edu (G.F.); 2Department of Macromolecular Science and Engineering, Virginia Polytechnic Institute and State University, Blacksburg, VA 24061, USA; vsamuel@vt.edu (S.V.); awhit@vt.edu (A.W.); 3Departments of Chemical Engineering and Materials Science and Engineering, Virginia Polytechnic Institute and State University, Blacksburg, VA 24061, USA; 4Center for Coastal Resources Management, Virginia Institute of Marine Science, Gloucester Point, VA 23062, USA; kirk@vims.edu; 5William & Mary Research Institute, William & Mary, Williamsburg, VA 23187, USA; jpmcde@wm.edu

**Keywords:** polyhydroxyalkanoate, poly(butylene succinate-*co*-adipate), ruminal fermentation

## Abstract

Using polyhydroxyalkanoate (PHA) materials for ruminal boluses could allow for longer sustained release of drugs and hormones that would reduce administration time and unneeded animal discomfort caused by continuous administration. The objective of this study was to determine ruminal degradability and kinetics of biodegradable polymers and blends. A proprietary PHA-based polymer, poly(butylene succinate-*co*-adipate) (PBSA), PBSA:PHA melt blends, and forage controls were incubated in rumen fluid for up to 240 h. Mass loss was measured after each incubation time, and digestion kinetic parameters were estimated. Thermogravimetric, differential scanning calorimetry, and intrinsic viscosity analyses were conducted on incubated samples. Generally, across treatments, mass loss was significant by 96 h with a minimum increase of 0.25% compared to 0 h but did not change thereafter. Degradation kinetics demonstrated that polymer treatments were still in the exponential degradation phase at 240 h with a maximum disappearance rate of 0.0031 %/h. Melting temperature increased, onset thermal degradation temperature decreased, and intrinsic viscosity decreased with incubation time, indicating structural changes to the polymers. Based on these preliminary findings, the first stage of degradation occurs within 24 h and PHA degrades slowly. However, further ruminal degradation studies of biodegradable polymers are warranted to elucidate maximum degradation and its characteristics.

## 1. Introduction

Polyhydroxyalkanoates (PHA) are aliphatic polyesters naturally produced via fermentative processes within various microorganisms during unbalanced growth when select nutrient sources are deprived in the presence of an abundant carbon source [1,2]. Polyhydroxyalkanoates have a vast array of applications due to their biodegradability in various environments, chemical diversity, insolubility in water, biocompatibility, and lack of toxins [2,3]. Despite these benefits, PHAs are quite expensive and tend to be mechanically brittle. Polymer blends can be developed by blending PHA with other biodegradable polymers. Polymer blends with polyhydroxyalkanoates can have improved mechanical properties that may make them more suitable for market and maintain their biodegradability [1]. Poly(butylene succinate-*co*-adipate) (PBSA) is a biodegradable polymer shown to improve elasticity and functionality of polymers while reducing costs when blended [4,5].

Polyhydroxyalkanoates, particularly poly(3-hydroxybutyrate) (PHB), have been of considerable interest in the biomedical field. Implanted devices such as screws, plates, films, and biodegradable sutures have been developed from these materials for procedures such as articular cartilage repair, meniscus repairs, and cardiovascular patch grafting. Disposable needles, syringes, sutures, surgical gloves, gowns, and more have been created from biodegradable polymers [2,6]. Due to their slow rate of degradation in tissues, PHA-based polymers have been introduced in pharmacology as a sustainable delivery method of anti-inflammatory drugs via microcapsules [7]. Despite the growing interest in these materials, PHAs are yet to be investigated for animal use, particularly in slow-release oral boluses.

Sustained-release boluses have many applications within ruminants; however, most boluses on the market only last three to five months and may require multiple dosages that may induce stress to the animal [8]. Using biodegradable materials such as PHAs to encapsulate drugs or hormones as a bolus may allow for longer sustained release of such materials that would reduce unneeded animal discomfort caused by repetitive administration. Depending on degradation rates and characteristics of PHAs or blends used, potential applications include but are not limited to anthelmintic treatments, estrous synchronization, growth supplements, and pre- and post-partum micronutrient boluses. Ruminant ability to degrade these materials within the rumen, as well as characterization of release, i.e., continuous or “burst” release of material, must be determined to assess suitability of these materials for intraruminal drug release.

Ruminants are herbivorous animals with a unique digestive tract and digestion process to accommodate their forage-based diets. The rumen, one of four compartments within the stomach, serves as a fermentation vat where most nutrient absorption occurs [9]. Ruminal fermentation is the process by which anaerobic and facultative microorganisms breakdown and ferment low-quality fibrous ingesta into utilizable energy sources for the animal, primarily volatile fatty acids [10]. The microbial consortium in the rumen consists of a wide diversity of organisms, with approximately 10^11^ cells/mL bacteria, 10^6^ cells/mL protozoa, and 10^4^ cells/mL fungi [9,10,11]. This may foster an environment capable of degrading PHA-based materials.

Biodegradable polymers such as PHA-based polymers and PBSA may be degraded abiotically or biotically. Abiotic factors that contribute to breakdown include mechanical stress, temperature, and chemicals. These cause minimal degradation by themselves but are synergistic to more extensive biotic degradation and may initiate the biodegradation process [12,13]. Polymers may degrade biotically via the formation of biofilms composed of a wide assortment of microbes including Gram-positive, Gram-negative, aerobic, facultative, and anaerobic bacteria. These microbes produce extracellular depolymerases that enzymatically degrade polymers by first cleaving polymer chains, which are then assimilated and converted to oligomeric and monomeric short chain fatty acids, carbon dioxide or methane, and water [6,14]. Some depolymerases are polymer specific, while others degrade a multitude of products [12]. Polymer degradation rates are highly variable and influenced by the aforementioned abiotic factors, microbial populations and concentrations, as well as the physical characteristics of the polymer such as crystallinity, molecular weight, density or thickness, and surface area [12,15,16,17].

Bacterial PHA depolymerases are commonly found from the *Cupriavidus, Alcaligenes, Comamonas,* and *Pseudomonas* genres [18]. *Pseudomonas* sp. are quite prevalent within ruminants, and along other cellulolytic bacteria are assumed to degrade PHA-based materials within the rumen [9,19,20]. Biodegradability of PBSA has been less widely evaluated, yet studies have demonstrated enzymatic degradation by *Roseateles depolymerans* [21], *Azospirillum* sp. [22], and *Leptothrix* sp. [23], among others. No studies to date, however, have evaluated the ability of the ruminal microbiome to produce these extracellular depolymerases. Because most known bacteria that produce depolymerases are within the phylum Proteobacteria, which comprise one of the more abundant phyla present in the rumen, ruminal microbes may have the ability to breakdown biodegradable polymer materials.

No prior research has specifically determined the ability of ruminants to degrade PHA-based materials and blends thereof within their gastrointestinal tracts. Estimating “true” digestive rates and the time needed to achieve maximum degradation of PHA materials and blends within the rumen is difficult due to the paucity of relevant literature and the large number of abiotic and biotic factors that influences those rates. The objective of this research was to assess digestibility kinetics of a proprietary PHA-based polymer and blends with PBSA via in vitro fermentation in rumen fluid. We hypothesized that biodegradable polymer materials would degrade in the ruminal environment, and that a blended material composed of PHA and PBSA would degrade at a faster rate than either material alone.

## 2. Materials and Methods

### 2.1. Animal Care and Use

The Institutional Animal Care and Use Committee of Virginia Tech approved all procedures involving dairy cows for collecting rumen contents (IACUC #18-229).

### 2.2. Sample Preparation for In Vitro Disappearance

Proprietary PHA-based polymer (Mirel P1004) nurdles produced by Metabolix, Inc. (Woburn, MA, USA) were purchased from Alterra Plastics (Clifton, NJ, USA). Poly(butylene succinate-*co*-adipate) (BioPBS^TM^) nurdles were purchased from Mitsubishi Chemical Performance Polymers (Greer, SC, USA). A poly(butylene succinate-*co*-adipate) (PBSA; Bionelle 3001MD; Showa Denko; Tokyo, Japan) and PHA (Mirel P1004) melt blend (90%wt PBSA, 10%wt PHA) in nurdle form as well as in a filament extrusion was developed. Briefly, PHA (Mirel P1004) and PBSA (Bionolle 3001MD; Showa Denko; Tokyo, Japan) were melt-blended and extruded using a pilot scale extruder at Alterra Plastics (Clifton, NJ, USA). A filament was developed from this blend at North Carolina State University.

To test the effect of filtration bag type on in vitro disappearance, PHA nurdle (PHA), poly(butylene succinate-*co*-adipate) nurdle (PBSA), PBSA:PHA melt blend nurdle (Blend), and PBSA:PHA melt blend filament (Filament) were inserted into porous F57 and R510 Dacron bags (25-micron and 50-micron porosity, respectively; (Ankom Technology Corp., Fairport, NY) using a polymer to bag surface ratio of approximately 10 mg/cm^2^. F57 bags were filled with 250 mg of sample and R510 bags were filled with 500 mg of sample. All bags were double sealed using an impulse heat sealer, ensuring the mass-to-area ratio was not reduced.

### 2.3. In Vitro Disappearance

Polymer samples were fermented in Daisy^II^ rotating jar in vitro incubators (Ankom Technology, Macedon, NY, USA). In vitro fermentations were conducted over 10 days using two incubators, each containing four fermentation jars. Duplicates of all polymer treatments in F57 and R510 ANKOM bags were fermented in a jar for each fermentation time point (0, 3, 6, 12, 24, 48, 96, and 240 h). In addition, duplicates containing corn silage standard, alfalfa hay standard, and blank bags of each type were included in each jar such that there were eight jars each filled with 28 bags, in vitro media, reducing agent, and inoculum as described below to maintain ruminal microbes throughout the study.

In vitro media and reducing agent were prepared as described by Goering and Van Soest [24]. Briefly, a 20 L buffer solution was prepared from ammonium bicarbonate and sodium bicarbonate in deionized water; a 20 L macro-mineral solution was prepared from sodium phosphate anhydrous, potassium phosphate anhydrous, and magnesium sulfate pentahydrate in deionized water; and a 100 mL micro-mineral solution was prepared from calcium chloride dihydrate, manganese chloride tetrahydrate, cobalt chloride hexahydrate, and ferrous chloride hexahydrate in deionized water. In vitro media (13.6 L) were prepared by mixing trypticase, deionized water, micro-mineral solution, buffer solution, macro-mineral solution, and resazurin (0.1% w/v) in that order. Reducing agent (735 mL) was prepared by mixing cysteine hydrochloride, deionized water, 1 N sodium hydroxide, and sodium sulfide, in that order.

On the day of fermentation, the prepared bags were placed in their respective fermentation jars with 1200 mL of media and warmed in a water bath at 39 °C under continuous purging with carbon dioxide to maintain the anaerobic environment of a rumen. A 4 L flask containing 3360 mL of media for blending with inoculum was also placed in a water bath at 39 °C and purged with carbon dioxide. Reducing solution was prepared, 60 mL was added to each fermentation jar, and 168 mL was added to the media for blending. A composited inoculum was prepared with rumen fluid and rumen solids collected from three cannulated, lactating Holstein cows.

The composited inoculum was prepared as follows. For each of the three cows, two 2 L thermos flasks were filled with approximately 1 L of rumen fluid and the remaining space was filled with rumen solids. Once in the laboratory, the top layer of surface solids exposed to the air was discarded from each thermos. The remaining mix of rumen fluid and rumen solids was strained through one layer of cheesecloth into a flask. Then, 550 mL of strained rumen fluid from each thermos was collected and strained through two layers of cheesecloth into the composite flask. From each thermos, approximately 280 g of strained solids was collected, mixed with 560 mL of preheated and reduced media, and then blended for 15 and 45 s at low and high speed, respectively (Waring blender HGB-300, Waring Commercial, New Hartford, CT, USA). The resulting blend was strained through two layers of cheesecloth into the composite flask. Every step was carried out under constant purging with carbon dioxide. After adding 800 mL of the composite inoculum to each fermentation jar, the jars were sealed with lids and placed into their designated incubators held at 39 °C. The average pH of inoculum was 7.06 ± 0.12 and the average temperature was 39.3 ± 0.5 °C.

After designated incubation times, jars were removed from the incubator. Bags were removed from the jars, rinsed by hand with ice water until the water ran clear, and dried in a forced-air oven for 24 h at 55 °C. Once fully dried, bags were weighed. Bags containing residues were corrected for the mass changes in respective blank bags during fermentation within incubation time to account for ruminal solids captured within bags. In vitro disappearance (IVD) was calculated according to Equation (1):(1)$$\mathrm{IVD}\left(\%\right)=\frac{\mathrm{Initial}\;\mathrm{Dry}\;\mathrm{Matter}\;\left(g\right)-\mathrm{Corrected}\;\mathrm{Undigested}\;\mathrm{Residue}\left(g\right)}{\mathrm{Initial}\;\mathrm{Dry}\;\mathrm{Matter}\left(g\right)}\times 100,$$

Digestion kinetic parameters for polymers and controls within the ruminal environment were estimated using the NLIN procedure in SAS according to the predicted digestibility Equation (2) [25]:Disappearance (%) = *A* + *B* × (1 − *e* ^(*−k* × *T*)^),(2)
where *T* is the time of incubation (h), *A* is the pool of immediately degraded material (%) at *T* = 0, *B* is the pool of potentially available material between 0 and 240 h (%), and *k* is the fractional rate of disappearance (%/h) of pool *B*. *B* was estimated to be (100 −
*C*
−
*A),* where *C* is the pool of non-digested material (%). For polymer treatments, *C* was estimated to be 0% because maximum degradation was not achieved during the study.

### 2.4. Thermochemical Analysis

Thermochemical analyses were conducted on samples incubated for 0, 24, 96, and 240 h in F57 bags. These time points were selected based on mass loss trends seen after analysis. Since bag type did not have an influence on mass loss, only samples incubated in F57 bags were selected for analysis. Thermogravimetric analysis (TGA) of fermented samples was conducted using a TA Instruments TGA Q50 (TA Instruments, New Castle, DE, USA). All samples were tested using aluminum pans in a nitrogen atmosphere. Samples were heated at 10 °C/min from room temperature to a maximum temperature of 450 °C. The onset thermal degradation temperatures (*T_d_*) were defined as the temperature at which 5% of the initial mass was lost from each degradation curve.

Differential scanning calorimetry (DSC) of fermented samples was performed using a TA Instruments DSC Q20. All samples were tested using aluminum pans in a nitrogen atmosphere. Samples of approximately 7.5–10.0 mg were first heated at 10 °C/min, cooled at 5 °C/min, and heated again using the heat/cool/heat method. PBSA was heated to a max heat of 150 °C and cooled to 20 °C; all other treatments were heated to 190 °C and cooled to 25 °C. The melting temperatures (*T_m_*) were determined from endothermic peaks in the first heating scan.

Intrinsic viscosity (η) was measured using an Ubbelohde viscometer. CHCl_3_ was selected as the solvent for analysis. In total, 15 mL of pure CHCl_3_ was added to the viscometer and pulled up to the feeder bulb through the capillary. The time was started when the meniscus crossed timing mark “A” and was stopped when the meniscus crossed timing mark “B”. Based on degradation trends observed from mass loss, *T_m_* and *T_d_*, serial dilutions were performed on PHA, PBSA, and PBSA:PHA in nurdle form fermented for 0 h, 24 h, and 240 h with concentrations of 5 mg/mL, 3.5 mg/mL, 2.5 mg/mL, 1 mg/mL, and 0.5 mg/mL. A volume of 15 mL was maintained between each concentration by removing the appropriate volume of polymer solution and replacing it with the same volume of CHCl_3_. Each concentration was run five times to ensure precision before the time was recorded. Relative viscosity (η*_rel_*) and specific viscosity (η*_sp_*) were calculated as follows:$$\eta_{rel}=\frac{t_{solution}}{t_{solvent}},$$
$$\eta_{sp}=\eta_{rel}-1;$$
such that $t_{solution}$ is the average time (s) it took for the polymer solution to pass from timing mark “A” to “B”, and $t_{solvent}$ the same for the pure solvent. The ratios of ln[η*_rel_*] to the polymer concentration and η*_sp_* to the polymer concentration were graphed against concentration, and η was determined as the intersection of the trendlines.

### 2.5. Statistical Analyses

Analyses were conducted separately to compare polymer compositions in nurdle form and to compare the PBSA:PHA melt blend in nurdle and filament formation. All statistical analyses were completed using the MIXED procedure in SAS. Polymer mass loss was analyzed with fixed effects of bag type, treatment, time, and interactions and random residual error. Disappearance kinetics parameters were analyzed with the fixed effects of bag type, treatment, the interaction, and random residual error. Some polymer treatments had two peaks within the first heating scan during differential scanning calorimetry due to multiple components; the initial peak was used for multiple comparisons of *T_m_*. The PBSA:PHA polymers had two degradation curves after thermogravimetric analysis was conducted; the first point of degradation was determined to be the PHA component and the second point of degradation was determined to be the PBSA component. Polymers *T_d_* and *T_m_* were analyzed with fixed effects of treatment, time, and the interaction and random residual error. All data were evaluated with an alpha value of 0.05.

## 3. Results

### 3.1. Mass Loss of Biodegradable Polymers

Bag type did not affect the mass loss of polymer nurdles nor the PBSA:PHA blend in nurdle and filament formation (*p* = 0.07; *p* = 0.95). Mass loss increased over time for each polymer treatment as expected, though significant mass loss was generally not achieved until 96 h of incubation and did not change thereafter (Figure 1). By 96 h, PHA achieved 0.25 ± 0.13%, PBSA 0.44 ± 0.16%, Blend 0.66 ± 0.20%, and Filament 0.72 ± 0.33% more mass loss than at 0 h.

Our polymer treatments did not differ from one another in nurdle form. Though, we observed a two-fold increase in mass loss in the Filament compared to the Blend (Figure 1, *p* < 0.0001). On average after 240 h of incubation, PHA had 0.53 ± 0.13%, PBSA had 0.49 ± 0.16%, Blend had 0.82 ± 0.20%, and Filament had 1.68 ± 0.33% total mass loss.

### 3.2. Degradation Kinetics

Ruminal digestibility kinetics parameters for polymer treatments and forage controls are shown in Table 1. The filament formation of PBSA:PHA had the greatest pool of immediately degraded material, nearly five times that of its nurdle counterpart (*p* = 0.02). The *A* fraction did not differ among polymers in nurdle composition (*p* = 0.50). Rates of degradation did not differ among the polymers in nurdle composition or among the PBSA:PHA blend in nurdle and filament composition (*p* = 0.38; *p* = 0.12). The maximum rate of disappearance achieved was 0.0031 %/h by the Blend, substantially less than what was achieved by the controls.

Predicted digestibility curves of polymers using the parameters in Table 1, along with alfalfa hay and corn silage controls, are shown in Figure 2. The alfalfa hay and corn silage controls demonstrated the full logarithmic curve typically seen with forages during ruminal degradation, reaching maximum degradation around 96 h indicated by a plateau. The polymer treatments, however, were still in the exponential phase of degradation indicated by a linear trend. When projected to 500 h of incubation, degradation was predicted to not exceed 2%. At this rate, it would take more than 30 months to approach 50% degradation of the Blend in the ruminal environment.

### 3.3. Thermochemical Analysis

The first melting temperature (*T_m_*) from the first heating scan of polymers after incubation for 0, 24, 96, and 240 h in rumen fluid are shown in Figure 3. Melting temperatures from the first heating scan were not different from the second scan. Two melting temperatures were observed in PHA as well as the PBSA:PHA blend in both nurdle and filament form, confirming the presence of more than one component in these polymers. However, two peaks were not present for the Blend or Filament until 24 h, and the Filament did not present a second peak at 96 h. Only the first peak temperature achieved within the first heating scan are shown for simplicity and comparison over time due to fermentation.

For PHA, two melting temperatures were clearly observed throughout the incubation period with significant increase in temperature by 96 h. When PBSA was added to the proprietary PHA-based polymer formulation, changes were observed in the Blend and Filament melting temperatures. For the Blend, two distinct *T_m_*s were not detected until after 96 and 240 h while for the Filament the two *T_m_*s were observed after 24 and 240 h. No significant increase in *T_m_* was observed for PBSA or Blend until 240 h. The Filament *T_m_* did not differ across incubation times.

Both the Blend and Filament presented two onset thermal degradation temperatures after 0, 24, 96, and 240 h as well. For assessment, these two temperatures were separated into the PHA component (Figure 4A) and PBSA component (Figure 4B). Interestingly, polymer *T_d_* had an opposite trend with incubation time. For the PHA components, the *T_d_* for PHA significantly decreased for 0, 24, and 96 h consecutively (269.55, 213.15, 144.60 °C, respectively). The *T_d_* associated with the PHA component within both the Blend and Filament polymers significantly decreased from 0 to 24 h by 119.50 ± 1.03 °C and 114.00 ± 1.60 °C, respectively. For the PBSA components, the *T_d_* for PBSA significantly decreased from 0, 24, to 96 h consecutively (360.45, 209.40, 199.60 °C, respectively). The *T_d_* associated with the PBSA component within both the Blend and Filament significantly decreased from 0 to 24 h by 137.00 ± 1.92 °C and 135.35 ± 0.81 °C, respectively.

### 3.4. Instrinsic Viscosity

The intrinsic viscosities of PHA, PBSA, and Blend after fermentation in rumen fluid for 0, 24, and 240 h in F57 bags are shown in Figure 5. Initially, the Blend had a viscosity 58% lower than PHA and PBSA; this trend held throughout fermentation, though this difference decreased to 32% after 240 h of fermentation. Viscosity decreased from 0 to 24 h by 40%, 18%, and 15% for PHA, PBSA, and Blend, respectively. The change in viscosity slowed with fermentation, only decreasing by 27%, 37%, and 9% from 24 to 240 h of fermentation for PHA, PBSA, and Blend, respectively.

## 4. Discussion

Repetitive administration of oral drugs due to their release times induces stress to the animal by additional handling. Other potential materials should be identified and studied to replace current slow-release bolus carriers that last only three to five months [8]. Polyhydroxyalkanoates are viable materials to investigate as they are biodegradable by several genera of bacteria [6,14]. Known bacteria that produce extracellular enzymes that hydrolyze these materials largely belong to the Proteobacteria phylum, which is abundant in the rumen [9,18,19,20,21,22,23]. With the abundant microbial community, elevated temperature, slightly acidic pH, and consistent motility in the rumen, the ruminal environment is presumed to foster degradation of PHA-based materials and blends with PBSA.

This is the first study to specifically evaluate the ability of a ruminal environment to breakdown a PHA-based polymer and PBSA. Proprietary PHA-based polymer nurdle, PBSA nurdle, and a melt blend of the two in nurdle and filament form underwent fermentation for up to 10 days in rumen fluid in Daisy^II^ Incubators. Mass loss of residues was determined and used to estimate ruminal digestive kinetics. Residues additionally underwent TGA and DSC to evaluate structural alterations to materials.

Previous literature has evaluated growth and health of monogastrics supplemented with PHA-based materials that indicated their degradation in the gastrointestinal tract. Specifically, aquaculture species supplemented with PHB, a particular PHA material composed of hydroxybutyrate monomers, had increased growth, decreased pH, and enrichment of the gut microbial community that indicated their degradation [14,26,27]. However, neither the PHA-based material utilized in this study nor PBSA and their blends’ mass loss and degradation kinetics in animal digestive tracts have been evaluated, and therefore close comparisons of degradation cannot be made at this time. Degradation studies of PHA polymers are mostly conducted with PHB and its copolymers in soil, compost, sewage sludge, or river water, which tend to be static environments with different microbial ecologies, temperature, and pH than the rumen environment. Degradation of biodegradable polymers is highly influenced by these factors, as well as the size, shape, and composition of the materials tested.

Although abiotic and biotic factors differ between these studies and the present one, the literature supports trends observed in our study in which multi-polymer blends degrade more readily than individual polymers. In a mass loss study over 35 days where PHB and copolymers were buried in ground soil at 28 °C, PHB homopolymer films had the slowest rate of degradation at 0.93 mg/d and achieved 10% degradation by 7 days whereas the fastest degrading copolymer with 4-hydroxybutyrate had a 75% increase in degradation rate at 1.63 mg/d and achieved 30% mass loss at 7 days [28]. This extreme increase in degradation compared to the present study in which we observed only 0.53% degradation of our PHA nurdle and 0.82% degradation of the melt blend with PBSA in nurdle formation after 10 days is likely attributed to structural form. The soil study utilized film discs that were approximately 30 mm in diameter and 0.035–0.045 mm thick [28]. Our study utilized commercial nurdles that were approximately 3 mm in diameter, significantly reducing the surface area-to-mass ratio.

Polymer formation may have a greater effect on degradation than composition. In a mass loss study in coastal waters over 160 days, films of PHB and a copolymer with 3-hydroxyvalerate had 58% and 54% degradation whereas their pellet counterparts had 38% and 13% degradation, respectively [29]. Additionally, the pellets had slower rates of degradation. Differences were explained by the pellets having a decreased surface area interface with the environment, slowing microbial attachment to the polymer surface and extending the lag phase of degradation [29]. This supports the two-fold increase in mass loss we observed with the Filament treatment compared to its nurdle counterpart in this study. Filament pieces were approximately 1 mm wide and 0.13 mm thick, allowing for an increased polymer/rumen fluid interface and bacterial attachment compared to the nurdles.

No studies have evaluated a PBSA:PHA blend for mass loss and degradation kinetics due to biodegradation, and generally studies with PBSA are limited. In one study, dog-bone shaped specimens of polybutylene succinate (PBS) and PBSA underwent biodegradation in soil and compost for 24 weeks [30]. Samples were also subjected to artificial weathering performed in cycles of exposure to UV light, elevated temperature, and artificial rainfall for up to 1800 h. When incubated in compost, PBSA mass loss was not observed until 4 weeks with 15% mass loss and it took PBS 12 weeks to achieve that same level of degradation. By 24 weeks, nearly 100% degradation of PBSA was achieved and PBS only achieved 70%. In soil, the biodegradation of PBSA and PBS was significantly less, and when subjected to artificial weathering, less than 1% mass loss was observed by 1800 h for both samples [30]. Together, these results indicate that copolymers of PBS degrade faster, and increased microbial concentrations accelerate degradation of these materials as they do PHAs. As we compare these degradation rates to PHA-based materials in similar conditions [28], PBSA degrades at a slower rate than PHA-based materials which has been supported by another study [13]. In our study, we observed that PBSA and PHA-based materials degrade at similar rates in the rumen environment; however, our study only lasted 10 days whereas former studies were conducted over a period of months and show deviations in degradation patterns that we may not be able to observe yet.

Trends seen within the present study after only 240 h support studies at longer time frames to capture additional polymer degradation kinetics in the rumen. Some studies have described biodegradable polymer degradation to have a lag phase requiring microbial films to develop on the surface of polymers followed by a two-phase curve of mass loss in which rapid degradation is preceded by slower degradation for more than 10 d [28,31]. At only 240 h of incubation, polymers in this study may still be in the first phase in which biofilms are developing and extracellular enzymes are beginning to hydrolyze polymer chains of the polymer surface. Although initial slow degradation will cause alterations to mechanical properties of the polymers, it may not cause significant mass loss. This could explain our observed trends with *T_d_* and *T_m_* within only 24 h and the minimal mass loss.

Melting temperature and onset thermal degradation temperature indicate structural integrity of polymers in the face of thermal challenge. Evaluating changes in these values over time indicates structural changes in polymers due to degradation. The observed increase in *T_m_* indicates the shortening of polymer chains and increasing crystallinity of polymers in this study likely caused by preferential depolymerase degradation of amorphous material [32,33]. Similarly, decreasing *T_d_* indicates a reduction in the thermal stability of the polymers, which may be explained by selective depolymerization of lower molecular weight polymer molecules in the amorphous region by 24 h [34]. The trending decrease in intrinsic viscosities of PHA, PBSA, and the Blend observed over the course of fermentation further supports the degradation of polymers within the first 24 h. Intrinsic viscosity is a reliable, indirect measure of a polymer’s molecular weight. Thus, changes in intrinsic viscosity over the course of degradation indicate alterations to the disposition of polymer molecules. The consistent decreased viscosity of the Blend compared to PHA and PBSA could be explained by chain scission that may have occurred during its preparation.

Previous studies support these findings in that PHA-based materials and blends thereof of various compositions show decreased molecular weight, increased melting temperatures, and decreased onset thermal degradation temperatures after environmental degradation by microbes [28,35]. Studies with PBSA indicate similar trends of increasing melting temperature and decreasing onset thermal degradation temperature due to enzymatic hydrolysis [13]. However, the present study does not control for abiotic hydrolysis; shortening of polymer chains within 24 h of incubation could be due to water-induced hydrolysis in the bulk of the polymer, as rumen fluid is mostly composed of water [13,36,37].

We propose that the slow, linear trend of degradation observed in this study is due to the beginnings of material surface erosion by bacteria [37]. While abiotic factors may contribute to fracturing and mechanical alterations of biodegradable polymers, only biotic factors will result in mass loss due to enzymatic hydrolysis and cleavage that reduces chains to low molecular weight oligomers, dimers, and monomers that are either released into the environment or taken up by the surrounding microbes [12,13,37]. Only one study has looked at the ability of microbes sampled from ruminal fluid to degrade PHA-based materials; no studies have been conducted with PBSA. Isolates of *Staphylococcus* sp. from rumen fluid collected from one mature cow were capable of degrading PHB as indicated by the development of clearing zones after plating serial dilutions of rumen fluid on PHB overlay plates [38]. Our study utilized a composited inoculum composed of rumen fluid collected from three mature, lactating dairy cows on the same diet and in the same housing conditions. Preliminary evidence from these two studies suggests some ruminants may contain microbes that can produce these depolymerases; however, a general conclusion that all ruminants can degrade all biodegradable polymers cannot be made at this time. Though a “core” microbiome containing 30 abundant bacterial groups was found in 90% of samples across a range of ruminant species, diets, and geographical regions, variation in bacterial diversity and prevalence still largely exists among animals primarily due to diet and housing [9,39]. Thus, the presence and abundance of depolymerase-producing bacteria in the rumen may vary among ruminants. Further incubation research with a larger sample size of ruminants under different housing and dietary conditions is needed to determine if the capability of intraruminal degradation of these materials is universal across ruminants.

Concurrently, in vivo studies must be conducted under these conditions to establish differences between in vitro and in vivo systems. It is possible that results seen in this study are undermined by the limitations of an in vitro system. Although the Daisy^II^ Incubator in vitro system is a widely accepted and reliable method for determining true ruminal digestibility of feedstuffs [40], it is a batch system with slight rotation. The live rumen is a continuous culture system with constant motility via contractions. In vivo, mechanical stress on polymers may be increased and accelerate abiotic fragmentation. This would allow more active bacterial formation of biofilms on larger surface areas and promote more rapid degradation and mass loss.

## 5. Conclusions

Based on these preliminary findings, PHA-based and PBSA materials are degradable in the rumen and may be a viable option for slow-release ruminal boluses. Within 24 h of incubation, polymer chains were likely cleaved via abiotic and biotic factors to alter mechanical structures of polymers as supported by thermochemical analyses. However, though polymer chains were shortening, mass loss was unsubstantial by 240 h of incubation, and the proprietary PHA-based polymer had the slowest rate of degradation. Further ruminal degradation studies of biodegradable polymers for a longer duration are warranted to elucidate full degradation kinetics and the relative contributions of water-induced and enzymatic hydrolysis to degradation. These studies should be conducted with a range of ruminant hosts fed different diets and in different housing conditions to determine the general ability of all ruminants to host depolymerase-producing microbes.

## Figures and Tables

**Figure 1 polymers-14-02103-f001:**
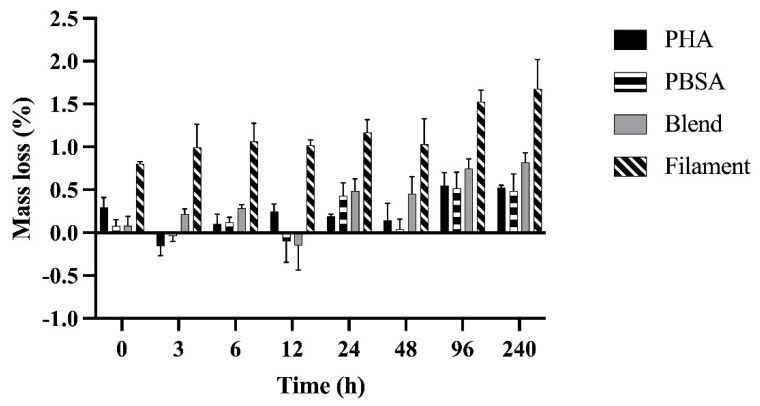
Mass loss (%) of polyhydroxyalkanoate nurdle (PHA), poly(butylene succinate-*co*-adipate) nurdle (PBSA), PBSA:PHA melt blend nurdle (90%wt PBSA, 10%wt PHA) (Blend), and a PBSA:PHA melt blend (90%wt PBSA, 10%wt PHA) filament (Filament) fermented in rumen fluid for up to 240 h. When comparing nurdles, bag type and all its interactions, as well as the interaction of treatment and time, were not significant. Treatment: *p* = 0.10. Time: *p* < 0.0001. When comparing Blend and Filament treatments, bag type and all its interactions, as well as the interaction of treatment and time, were not significant. Treatment: *p* < 0.0001. Time: *p* = 0.04. Data shown are least squared means of polymers in F57 and R510 bags in duplicate (LSM ± SEM).

**Figure 2 polymers-14-02103-f002:**
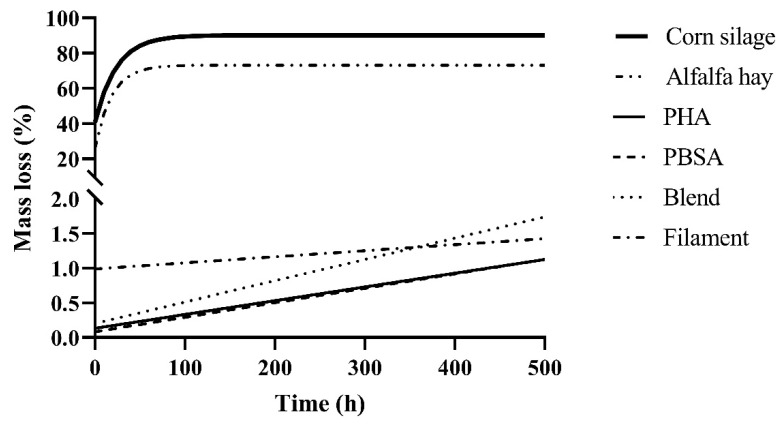
Projected degradation of forage controls alfalfa hay and corn silage and polymer treatments PHA, PBSA, Blend, and Filament. Projected degradation was determined from calculated disappearance kinetics parameters *A* and *k* as determined from the potential degradability equation (Table 1), where *A* is the immediate disappearance (%) at *T* = 0, and *k* is the fractional disappearance rate (%/h) of available material between *T* = 0 and *T* = 240. Kinetics parameters plotted for polymer treatments are mean values of those fermented in F57 and R510 bags in duplicate. Kinetics parameters plotted for forage controls are mean values of those fermented in only F57 bags in duplicate as bag type was significant.

**Figure 3 polymers-14-02103-f003:**
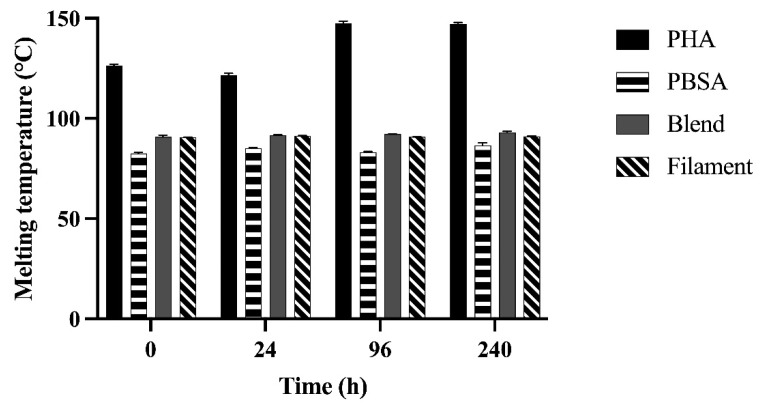
Melting temperature of PHA, PBSA, Blend, and Filament treatments after fermentation in rumen fluid for 0, 24, 96, and 240 h in F57 bags. Melting temperature (°C) of the treatments was determined from the first endothermic peak in the first heating scan. When comparing nurdles, Treatment: *p* < 0.01, Time: *p* < 0.01, Treatment × Time: *p* < 0.01. When comparing Blend and Filament treatments, Treatment: *p* = 0.01, Time: *p* = 0.05, Treatment × Time: *p* = 0.23.

**Figure 4 polymers-14-02103-f004:**
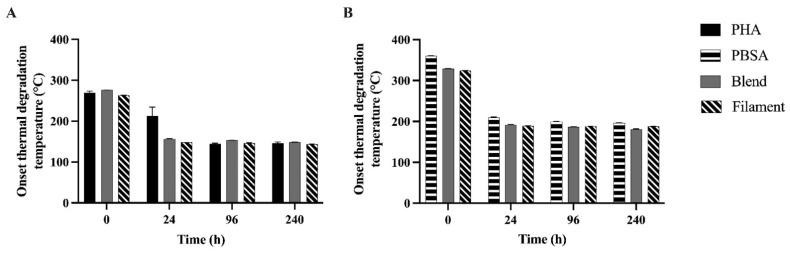
Onset degradation temperature of PHA, PBSA, Blend, and Filament treatments after fermentation in rumen fluid for 0, 24, 96, and 240 h in F57 bags separated by: (**A**) PHA component and (**B**) PBSA component. Onset thermal degradation temperature (°C) of treatments were determined to be the temperature at which 5% of the initial mass degraded. For the PHA components, time was the only significant parameter. Time: *p* < 0.0001. For the PBSA components: Treatment: *p* < 0.0001, Time: *p* < 0.0001, Treatment × Time: *p* < 0.0001. Data are shown as least squared means of treatments in duplicate (LSM ± SEM).

**Figure 5 polymers-14-02103-f005:**
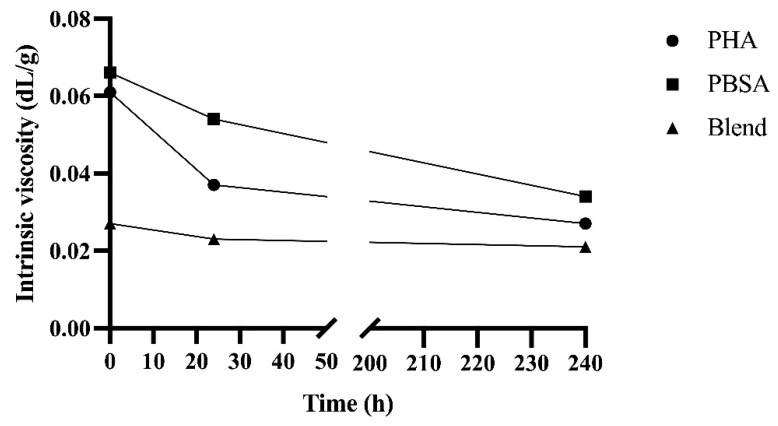
Intrinsic viscosities of PHA, PBSA, and Blend treatments after fermentation in rumen fluid for 0, 24, and 240 h in F57 bags.

**Table 1 polymers-14-02103-t001:** Ruminal disappearance kinetics parameters of alfalfa hay and corn silage forage controls and PHA, PBSA, Blend, and Filament.

							Composition ^3^		Formation ^4^	
Parameter	Alfalfa	Corn	PHA	PBSA	Blend	Filament	SEM	*p*-Value	SEM	*p*-Value
^1^*A*, %	43.26	58.18	0.13	0.08	0.20 ^b^	0.99 ^a^	0.07	0.50	0.14	0.02
^2^*k*, %/h	12.27	4.93	2.00 × 10^−3^	2.10 × 10^−3^	2.10 × 10^−3^	8.85 × 10^−4^	6.17 × 10^−6^	0.38	8.25 × 10^−6^	0.12

^1.^*A* = pool of immediately degraded material at *T* = 0. ^2.^
*k* = fractional rate of disappearance of the pool of potentially available material. ^3.^ Comparison of PHA, PBSA, and Blend treatments. ^4.^ Comparison of Blend and Filament treatments. Data are shown as least squared means of treatments incubated in both R510 and F57 bags in duplicate. Different letter superscripts in a row indicate significant differences between polymers (*p* < 0.05). Forage controls were not included in statistical analysis as only polymer degradation was of interest. Bag type nor its interactions were significant for neither the nurdles or two formations of PBSA:PHA blend.

## Data Availability

The datasets presented in this study are openly available in Open Science Framework at https://doi.org/10.17605/OSF.IO/NW5FA.

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
