# Peer review of "Digestibility Kinetics of Polyhydroxyalkanoate and Poly(butylene succinate-co-adipate) after In Vitro Fermentation in Rumen Fluid"

_polymers, 2022, doi:10.3390/polym14102103_

Round 1
Reviewer 1 Report
The aim of the study was to assess digestibility kinetics of the blend and its components, in the in vitro fermentation in rumen fluid. Unfortunately, the goal set by the authors is not consistent with the research conducted, probably due to the fact that the researchers were not polymer chemists but biologists. Homopolymers have not been studied so it cannot be said how quickly they will degrade. The studies also lack molar mass determinations that could be helpful in interpreting the results. The very assumption of the work is interesting, but the way it is presented is unacceptable.
- “polybutylene succinate adipate”: should be “poly(butylene succinate-co-adipate)”. The IUPAC rules should be used for polymer names. Please correct throughout the text.
- “24 hours”: Please use SI symbols. Please correct throughout the text.
- Please avoid the statement “copolymer blend”, especially in the case where the authors did not provide the characteristics of the PHA used.
- Where the authors refer to the tested blend (for example abstract), they cannot use the name "biopolymer" as it refers only to natural polymers (see IUPAC recommendations). Polymers obtained in the reactor, even by a reaction catalyzed by microorganisms, are no longer biopolymers. Biobased can be used for the blend and polymers. Microbial polymer can be also use for the PHA itself.
- The fragment from lines 40 to 44 is incomprehensible. PHA is not one polymer, but a family of polymers, including copolymers, but not only. Accordingly, PHAs differ in properties, including crystallinity. There is no such thing as a "PHA monomer", has to be a “constitutional unit”. Please correct throughout the text. And if we get PHA from the monomer, it is a polymerization reaction and the resulting polymer will be synthetic, not bacterial.
- “Polybutylene succinate adipate (PBSA) is a bio-based plastic”: Here authors are talking about polymer, not plastic (material made from a wide range of polymers).
- Please explain the abbreviations where they are first used (Figure 1).
- I think the presentation of DSC and TGA is too simplistic. There are two copolymers in the blend, i.e. there should be 4 polymers (4 signals) (if not, then it should be properly commented). In addition, after degradation, the components of the copolymer degrade at different rates, so choosing one is unreliable. It should be presented more comprehensively.
- The choice of PHB (should be PHBV) for comparison with the authors' research is also incomprehensible. PHB copolymers differ significantly in properties from PHB itself, as they are composed of different constitutional units.
Reviewer 2 Report
The manuscript describes the mass loss kinetics of polyhydroxyalkanoate and polybutylene succinate adipate after in vitro fermentation in rumen fluid. The obtained data are valuable in this scientific area. This manuscript can be accepted. But there are some serious issues that need revision.
- What are the initial molecular weights of polyhydroxyalkanoate and polybutylene succinate adipate? Mw is almost the most important parameter of polymers.
- What monomer content of polyhydroxyalkanoate? Why do the authors use such a common chemical name for a class of polymers? Poly(3-hydroxybutyrate), polylactic acid, and poly-ε-caprolactone are all polyhydroxyalkanoates.
- Why did the authors not investigate the kinetics of molecular weight change of polymers? After all, the main mathematical models of biodegradation kinetics are based precisely on the analysis of the Mw decrease. Without this, the data cannot be of sufficient value.
- «The melting temperatures (Tm) were determined from endothermic peaks in the first heating scan.» Why did you use the first heating scan instead of the second scan?
- The article focuses on the development of a product for the controlled release of substances in the stomach. The release of substances can have a different mechanism - diffusion and decomposition of the shell. It is important to understand how the desired substance will be released. Due to the shape of the shell, the polymer may have different degradation kinetics than granules and filaments. Could the authors explain why did not they make a bolus from these materials?
- «Filament pieces were approximately 1 mm wide and 0.13 mm thick, allowing for an increased polymer/rumen fluid interface and bacterial attachment compared to the nurdles». Why did you use filaments? If you just want to achieve faster degradation, the filaments could be turned into powder. For a correct comparison, it is necessary to take the same size of objects.
- Also, it was not mentioned about the pH of the solutions, but it is an important parameter for the degradation of polymers and also for gastric microbiota.
Round 2
Reviewer 1 Report
The presentation of the entire work is still unacceptable. There are too many factual errors. I would like to ask the authors to eliminate mistakes that should not be found in the manuscript for Polymers.
1. “Digestibility kinetics of polyhydroxyalkanoate blend and poly(butylene succinate-co- adipate) after in vitro fermentation in rumen fluid”
Please dont use for PHA " PHA blend" wording.
Mirel P1004 is "based on polyhydroxyalkanoate polymers (PHAs) and are made by fermentation using renewable carbon-based feedstocks, making them 100% biobased in neat form"
If the authors did not analyze or cite the Mirel P1004 analysis by other authors, it cannot be say whether it is a copolymer, blend or copolymer/polymer blend.
2. PHA is not commonly referred to as "bio-based". Usually it is refer to as a "biodegradable" polymer.
It is important to be aware of what particular terms mean and not to replace them in an ill-considered way.
As PHA and PBSA are polymers of different origins: PHA is generally all derived from biomass and PBSA may or may not be bio-based, I would use "biodegradable" rather than bio-based (which may not necessarily be biodegradable).
However, when discussing polymers, it is not necessary to keep repeating that they are bio-based or biodegradable.
For example "Bio-based polymers can be degraded abiotically or biotically." This is not true as bio-based polymers may not degrade such as bio-based PP or PE.
“Polyesters” or the “used/investigated/described polymers” would be used here.
3. Line 46: “developed by blending PHA constitutional units with other biodegradable polymers”; In this case, it is enough to say: “developed by blending PHA with other biodegradable polymers”. Neither the monomer nor the constitutional unit is applicable here.
4. Line 286: One "biopolymers" escaped the authors' attention. Please correct.
Reviewer 2 Report
It is totaly incorrect to study the biodegradation of a polymer without knowing its initial chemical structure, molecular weight, and content (for blends). The authors could not provide an answer to this key question. The authors gave only partial answers to other questions.
Author Response
We greatly appreciate your time and consideration of our manuscript. As we described previously, we are using proprietary polymers (Mirel P1004, BioPBS, and Bionelle 3001). Legally, we cannot determine the exact composition of the polymers ourselves. Therefore, we, unfortunately, cannot provide this information.
As a preliminary investigation into how these types of biodegradable materials degrade within rumen fluid, we determined that reporting changes in mass loss, intrinsic viscosity, melting temperature, and onset thermal degradation temperature from their initial values (0 h) up to 240 h to be adequate measures to determine these materials' degradability. At this initial point of our study, we were not specifically interested in the individual components and their kinetics, and thus we used proprietary materials available to us. The results propel us to conduct further studies with polymers to be developed by us so that we can properly provide their specific composition and evaluate their potential for ruminal drug release products.
We sincerely hope this does not completely impede this manuscript from further consideration, but unfortunately, we can not provide the requested information at this time.
Round 3
Reviewer 1 Report
OK. Thank you
Author Response
We greatly appreciate your time and consideration of our manuscript. We have not made any further edits.